# Prevalence and Factors Associated with HIV Testing Among Men Aged 15–54 Years in Kenya—Evidence from the 2022 Demographic and Health Survey

**DOI:** 10.3390/ijerph22081291

**Published:** 2025-08-18

**Authors:** Ipeleng Caroline Victoria Melato, Alfred Musekiwa, Siphesihle Robin Nxele

**Affiliations:** School of Health Systems and Public Health, Faculty of Health Sciences, University Of Pretoria, Pretoria Private Bag X20, Hatfield, Pretoria 0028, South Africa; alfred.musekiwa@up.ac.za (A.M.); sr.nxele@up.ac.za (S.R.N.)

**Keywords:** sub-Saharan Africa, Kenya, HIV testing, men, Kenya demographic health survey, 15–54 years

## Abstract

Sub-Saharan Africa bears the heaviest burden of HIV/AIDS. Kenya alone has an estimated 1.4 million people living with HIV. Therefore, this study determined HIV testing prevalence and associated factors among men aged 15–54 years in Kenya. The study is a secondary data analysis of the 2022 Kenya Demographic and Health Survey, a nationally representative population-based cross-sectional survey. A multivariable logistic regression model was used to determine factors associated with HIV testing. Survey weights were used to adjust analyses for unequal sampling probabilities. Out of 14,453 men included in the study, the prevalence of self-reported HIV testing was 73.5%, which was higher among men aged 30–34 years old compared to the 15–19 years (90.2% vs. 33.3%), married or living with a partner (89.1% vs. 55.5%), residing in urban areas (82.5% vs. 67.8%), with higher education (90.6% vs. 58.4%), employed (80.5% vs. 43.1%), richest (83.8% vs. 60.4%), and those with three or more sexual partners (81.7% vs. 68.0%) groups. Targeted interventions to encourage more men to participate in regular HIV testing are needed. This can be achieved by bringing HIV testing sites closer to males through HIV self-testing and community testing, particularly home-based testing.

## 1. Introduction

HIV/AIDS remains a major global public health problem [1], with 39 million people living with HIV, 17.4 million of them being men aged 15 and older, and 630,000 dying of HIV-related illness worldwide in 2022. However, in 2018, only 190 million people tested for HIV and received their results [2].

Kenya has the seventh highest rate of HIV in the world [3]. Despite significant progress over the past 30 years, HIV remains a major public health concern in Kenya, contributing substantially to the country’s overall disease burden. The national HIV prevalence stands at 3.7% [4], with a notable gender disparity of 2.16% among males and 4.46% among females [5].

The high prevalence of HIV and AIDS has been linked to limited knowledge of transmission pathways, negative attitudes toward condom use, and engagement in high-risk sexual intercourse and multiple partnerships [6]. 

HIV testing and counselling (HTC) services are a public health program designed to identify new HIV cases and reduce the spread of the virus, thereby helping to eradicate the HIV epidemic. It is, therefore, crucial to increase HIV testing uptake by making it more accessible to communities where it has not been previously available [7]. Despite such programs in Kenya, in 2022, HIV testing uptake was 85% for women and 73% for males, with varying rates by county.

Several factors, including age, wealth index, region of residence, marital status, level of education, use of condoms, exposure to the media, and age at first sex have been shown to be significantly associated with HIV testing among men [7,8,9,10]. Higher rates of HIV testing have been found in married people and lower rates of HIV testing have also been found in never-married males aged 15–24 years [11]. A study among Ghananian men found that HIV testing was higher in urban compared to rural residents (19.9% vs. 10.9%, respectively) [12]. With regard to the socioeconomic factors, those with no education or lower attainment had a decreased chance of HIV testing [13]. Individuals with comprehensive knowledge about HIV/AIDS were more likely to test compared to those with no HIV/AIDS knowledge [14]. In addition, employment status and HIV testing uptake has an association. Employed individuals were more likely to test than those who were not [15]. Behavioral factors included risky sexual behaviors; studies indicate that there is no association between HIV testing and individuals with a history of sexually transmitted infections (STIs) [16].

Kenya has introduced a national plan to address low HIV testing rates by implementing HIV testing in two main settings, community-based and facility-based. The community-based approach targets key groups, such as orphans, vulnerable children, adolescents, youth, and workplace populations. The facility-based approach integrates HIV testing into all inpatient and outpatient services. Additional strategies include HIV self-testing, index testing, social network strategies, and voluntary counselling and testing [17].

UNAIDS introduced 95-95-95 targets for HIV testing, ART initiation, and viral load suppression, respectively. The first 95 is crucial as it is the gateway to HIV services for either prevention or treatment. It is important that those who are unaware of their HIV status be tested so that if they are HIV positive, they can initiate treatment and, therefore, not continue to spread the virus. Improving HIV testing uptake is essential for the accomplishment of Sustainable Development Goal 3 (SDG 3), which aims to end epidemics of AIDS, tuberculosis (TB), malaria, and other infectious diseases by 2030. It is concerning that HIV testing in males is lower than in females, and this continues to be a problem globally. While there is a substantial body of scientific literature on HIV testing and its associated factors, much of this research tends to focus on pregnant women, adolescent girls and young women, and key populations, such as sex workers and men who have sex with men. Although studies focusing on men do exist, particularly in relation to specific subgroups or comparative analyses, they are relatively fewer in number compared to those on women and other key populations. Therefore, more male-specific research is needed to address the persistent gender gap in HIV testing uptake and outcomes. This study aimed to determine the prevalence and factors associated with self-reported HIV testing among men aged 15–54 years in Kenya.

## 2. Methods

### 2.1. Study Design and Setting

The study was a quantitative secondary data analysis of Kenya’s 2022 Demographic Health Survey (KDHS) data set, a nationally representative population-based cross-sectional survey. The study was conducted in Kenya, a country with the seventh highest HIV cases in the world [3]. The 2022 KDHS was carried out in 47 counties of Kenya, including rural and urban areas.

### 2.2. Study Population and Sampling

The study population included males aged 15–54 years who lived in Kenya at the time of the survey. In the survey, stratified, two-stage cluster random sampling of households was conducted. For the 2022 KDHS, the sample was drawn from the Kenya Household Master Sample Frame (K-HMSF), and 42,022 households were sampled for the 2022 survey. The survey was carried out by the Kenya National Bureau of Statistics (KNBS), in coordination with the Ministry of Health (MoH) and other stakeholders. For further information on sampling, the details are available in the 2022 Kenya DHS report [18].

Survey weights were used to account for the survey design (weights, stratification, and clustering) since the 2022 Kenya DHS survey used the two-stage stratified sample design.

### 2.3. Measurements

#### 2.3.1. Study Outcome

The primary outcome of interest was self-reported HIV testing; participants were asked if they had ever tested for HIV, with a yes/no response.

#### 2.3.2. Independent Variables

The potential factors associated with self-reported HIV testing include age in years, marital status, place of residence, education level, employment status, wealth index, comprehensive HIV knowledge, condom use, STI status, and number of sexual partners.

#### 2.3.3. Data Management

The study used survey data, which were accessed through the DHS program after obtaining necessary approvals for using the data. The data were provided in a STATA dataset (.dta) file, which was then imported into STATA MP 14 for analysis. To ensure quality data, evaluation of the overall quality of the data was conducted, and missing values were checked using STATA.

For data security and confidentiality, the datasets were securely stored, access was limited to the authorized researchers, and password protection was used. The data were anonymized to ensure confidentiality.

#### 2.3.4. Ethical Considerations

Ethical clearance for this study was obtained from the University of Pretoria Faculty of Health Sciences Research Ethics (ethics reference number 380/2024).

#### 2.3.5. Statistical Analyses

The results of the study were analyzed using STATA MP software, version 14. Descriptive statistics were used to summarize the characteristics of participants. Bivariate and multivariable logistic regression models were used to determine demographic, socioeconomic, and behavioral factors associated with self-reported HIV testing. Backward elimination was used in building the final multivariable model from the bivariate analyses, where factors with *p*-value < 0.05 were shortlisted as candidates. A *p*-value ≤ 0.05 was used to indicate statistical significance. Unadjusted odds ratios (ORs) and adjusted odds ratios (aORs), with their corresponding 95% confidence intervals (CIs) and *p*-values, were presented to highlight the strength and direction of the association. All analyses were adjusted for unequal selection probabilities using survey weights.

## 3. Results

The 2022 Kenya Demographic and Health Survey had a sample size of 14,453 males aged 15–54 years who agreed to participate in the study and answered relevant questions on HIV testing.

### 3.1. Demographic Characteristics

Of the 14,453 men analyzed, 22.0% were aged 15–19 years and 5.5% were aged 50–54 years. About half of the males were married or living together with their partners (48.1%), and 61.1% resided in rural areas (Table 1).

### 3.2. Prevalence of Self-Reported HIV Testing

Among the sampled men aged 15–54 years, the overall prevalence of self-reported HIV testing (ever tested for HIV) was 73.5% (95% CI 72.4–74.6).

### 3.3. Factors Associated with Self-Reported HIV Testing

#### 3.3.1. Demographic Factors

In adjusted analyses, the odds of self-reported HIV testing were higher in the males aged 30–34 years than in the adolescent boys aged 15–19 years (90.2% vs. 33.3%; adjusted odds ratio [aORs] 4.81; 95% CI 3.51–6.59). Males who were married or living together with their partners had higher odds of self-reported HIV testing than those who were not in a union (89.1% vs. 55.5%; aOR 2.36; 95% CI 1.87–2.98). Men who resided in the urban areas had higher odds of self-reported HIV testing than in rural areas (82.5% vs. 67.8%; aOR 1.41; 95% CI 1.12–1.79). (Table 2).

#### 3.3.2. Socioeconomic Factors

In adjusted analyses, HIV testing was higher in men with higher education than those with no education (90.6% vs. 58.4%; aOR 5.48; 95%; CI 3.70–8.11). Employed men had higher odds of self-reported HIV testing than those who were not employed (80.5% vs. 43.1%; aOR 1.37,95% CI 1.08–1.72). When looking at the wealth index, the odds of self-reported HIV testing were higher in the richest group than those who were from the poorest (83.8% vs. 60.4%; aOR 1.41, 95% CI 1.00–2.12) (Table 2).

#### 3.3.3. Risky Sexual Behaviors

Comprehensive HIV knowledge was not significantly associated with self-reported HIV testing in both unadjusted and adjusted analyses. Males who had an STI in the last 12 months had higher odds of self-reported HIV testing (83.5% vs. 73.2%; OR 1.85; 95% CI 1.38–2.49); however, this was no longer significant after adjusting for confounders. Men who used condoms during their last sex had lower odds of self-reported HIV testing than those who did not use condoms (79.3% vs. 86.2%; OR 0.61; 95% CI 0.53–0.71); however, this was no longer significant after adjusting for confounders. Males with three or more partners had higher odds of self-reported HIV testing than males with one partner (85.1% vs. 68.0%; OR 2.67; 95% CI 2.29–3.11), even after adjusting for confounders (aOR 1.41, 95% CI 1.17–1.70) (Table 2).

In the Type III analysis, most variables remained statistically significant; however, the associations with the “poorer” wealth quintile and having “two or more sexual partners” were no longer statistically significant (Appendix A Table A1).

## 4. Discussion

This study found that, in 2022, about three-quarters of 15–54-year-old Kenyan men self-reported that they had ever tested for HIV, thus identifying an unmet need for HIV testing among men in Kenya. Adolescent boys aged 15–19 years had the lowest self-reported HIV testing uptake, with about 1 in 3 people as compared to men aged 30–34 years, where about 9 in 10 people reported HIV testing, similar to another study conducted in sub-Saharan Africa [15]. The reason for this could be that adolescents are largely dependent on their parents for decision-making authority to undergo HIV testing. Furthermore, their unwillingness to go for HIV testing may be due to stigma, fear, or anxiety.

This study found that married men or those living with a partner were more likely to test for HIV than the ones who were not in a union, separated, widowed, or divorced. This was in line with a study conducted in East Africa [7]. The common practice of mandatory premarital HIV testing in many sub-Saharan African countries may help explain this finding. Previous studies have shown that HIV testing is often a prerequisite for marriage in certain countries, such as Nigeria, Ethiopia, and Cameroon [19].

The study found the uptake of HIV testing to be lower in rural than in urban areas, which is consistent with a study conducted in South Africa [13], which could be linked to inadequate resources and structural barriers to health care with regard to geographical and financial accessibility.

Consistent with a previous study [9,20], this study found that those with no formal education had significantly lower odds of self-reported HIV testing compared to those with higher levels of education. Education and HIV-related knowledge are well established predictors of testing behavior, as those with more education may have greater awareness of HIV risks, better access to information, and improved ability to navigate health services.

In this study, men who were employed had higher odds of HIV testing than those who were unemployed, which is consistent with a previous South African study [21]. The possible reason for this could be the implementation of workplace interventions, as they have proven to be an essential pathway for HIV testing, especially where there are high incident rates and limited access to care. Additionally, worksite HIV testing programs have also increased convenience for HIV testing among difficult-to-reach populations [22]. Similar to a previous study conducted in South Africa [23], the current study found that men in the richest wealth index had higher odds of self-reported HIV testing than those from the poorest index. Despite HIV testing being free in Kenya’s government institutions, transportation costs may limit access to and from health facilities for those who are poor.

This study found that men with three or more sexual partners were more likely to test for HIV than those with one sexual partner. This could be because those with multiple partners are aware that they may be more vulnerable to HIV, which increases their desire to seek HIV testing. Surprisingly, this study found no significant association between HIV/AIDS comprehensive knowledge and self-reported HIV testing. This finding highlights a discrepancy when compared to the results of previous studies [14,19]. The reason for this could be lack of formal education, challenges in processing large amounts of information, and limited access to information and services with appropriate and updated information. Having comprehensive HIV knowledge is generally acknowledged to be an initial step in HIV testing, as it changes risky sexual behaviors in at-risk populations and also assists with the acceptance of ART [24].

The findings of this study, based on the 2022 KDHS, indicate a notable improvement in HIV testing uptake among Kenyan men aged 15–54 years, with 73.5% reporting ever having tested. This represents a marked increase from the 2008–09 KDHS estimate of approximately 40% [25], and is a continued rise from the 2012 KAIS estimate of 55% [26], reflecting Kenya’s sustained efforts to scale up testing services through strategies, such as provider-initiated testing and counselling (PITC), HIV self-testing, and community-based outreach.

These findings align with regional evidence demonstrating improved HIV testing among men following targeted interventions. For example, studies conducted in Ethiopia [7] and Senegal [27] similarly reported enhanced uptake associated with male-friendly strategies and decentralization of services. Furthermore, evidence from a sub-Saharan African study [19] highlighted the role of reducing stigma and improving partner communication in enhancing testing rates, consistent with Kenya’s national HIV response. Nonetheless, this study also reveals persistent disparities, particularly among rural men, those with limited education, and younger individuals, indicating the need for targeted, equity-focused interventions to ensure inclusive progress in HIV testing uptake.

## 5. Strengths and Limitations

The main strength of the study was its use of secondary data from the DHS, which provides a high-quality survey and a large sample of nationally representative data of 15–54-year-old Kenyan men. However, the study had some limitations. It relied on existing variables and did not explore new ones related to HIV testing, such as the impact of health workers, stigma or discrimination among males, and individual testing barriers. The study revealed factors associated with self-reported HIV testing that can inform targeted interventions. The use of self-reported responses could lead to recall and social desirability biases, which may compromise the findings, though this effect is minimal due to the large sample. Being cross-sectional, the study cannot determine a causal link between risk factors and HIV testing. Some independent variables, like condom use and number of sexual partners, had missing data; however, they were not statistically significant and may not have affected the results of multivariate analysis.

## 6. Conclusions

The study findings indicate an unmet need in HIV testing among 15–54 years old Kenyan men, especially among adolescent boys aged 15–19 years old. An age of 30–34 years, marriage, urban areas, employment, a higher level of education, and wealth index were significantly associated with HIV testing. Increasing HIV testing among males is crucial for eliminating the HIV/AIDS pandemic and meeting the UNAIDS 95-95-95 targets.

## 7. Recommendations

To improve HIV testing among adolescent boys in Kenya and support the UNAIDS 95-95-95 targets, it is crucial to enhance awareness and HIV knowledge, integrate regular HIV testing in schools, and implement public health interventions, such as community health posts and information centers. Expanding access to rapid tests, self-testing kits, and mobile health vans can further increase HIV testing uptake among men. Additionally, qualitative studies are recommended to explore barriers to HIV testing.

## Figures and Tables

**Table 1 ijerph-22-01291-t001:** Demographic characteristics of the males in the Kenya Demographic and Health Survey 2022 (N = 14,453).

Factors	*n*	%
Age in years, mean	29.9	
15–19	3349	22.0
20–24	2332	16.6
25–29	2109	15.7
30–34	1748	12.4
35–39	1628	10.9
40–44	1386	9.2
45–49	1117	7.7
50–54	784	5.5
Marital status *		
Never in union	6508	45.6
Married or living together	7067	48.1
Separated/widowed/divorced	878	6.3
Residence		
Urban	5232	38.9
Rural	9221	61.1

* some data were missing.

**Table 2 ijerph-22-01291-t002:** Factors associated with HIV testing among 15–54 males in the 2022 Kenya Demographic and Health Survey (N = 14,543).

			Bivariate			Multivariate		
Variable	Categories	%HIV Tested	OR	95% CI	*p*-Value	aOR	95% CI	*p*-Value
Demographic factors								
Age group (years)	15–19	33.3	Ref	-	-	-	-	-
	20–24	73.2	5.49	4.66–6.46	<0.001 *	2.54	2.05–3.14	<0.001 *
	25–29	89.4	17.0	14.03–20.60	<0.001 *	4.7	3.59–6.18	<0.001 *
	30–34	90.2	18.53	14.88–23.07	<0.001 *	4.81	3.51–6.59	<0.001 *
	35–39	88.9	16.03	12.94–19.87	<0.001 *	4.26	3.09–5.87	<0.001 *
	40–44	85.9	12.25	9.92–15.12	<0.001 *	3.31	2.35–4.67	<0.001 *
	45–49	84.2	10.68	8.39–13.61	<0.001 *	2.67	1.86–3.84	<0.001 *
	50–54	86.1	12.46	9.48–16.36	<0.001 *	3.09	2.13–4.48	<0.001 *
Marital status	Not in union	55.5	Ref	-	-	-	-	-
	Married or living together	89.1	6.57	5.78–7.48	<0.001 *	2.36	1.87–2.98	<0.001 *
	Separated, widowed, or divorced	84.9	4.50	3.56–5.70	<0.001 *	1.75	1.29–2.35	<0.001 *
Residence	Rural	67.8	Ref	-	-	-	-	-
	Urban	82.5	2.23	1.95–2.55	<0.001 *	1.41	1.12–1.79	0.004 *
Socioeconomic factors								
Education level	No education	58.4	Ref	-	-	-	-	-
	Primary	68.4	1.54	1.22–1.95	<0.001 *	2.31	1.71–3.13	<0.001 *
	Secondary	70.6	1.72	1.35–2.18	<0.001 *	3.85	2.76–5.39	<0.001 *
	Higher	90.6	6.87	5.04–9.37	<0.001 *	5.48	3.70–8.11	<0.001 *
Employment	Not employed	43.1	Ref	-	-	-	-	-
	Employed	80.5	5.45	4.78–6.22	<0.001 *	1.37	1.08–1.72	0.008 *
Wealth	Poorest	60.4	Ref	-	-	-	-	-
	Poorer	66.8	1.32	1.15–1.51	<0.001 *	1.19	0.99–1.42	0.065
	Middle	70.1	1.54	1.34–1.78	<0.001 *	1.23	1.01–1.49	0.038 *
	Richer	80.8	1.77	2.33–3.28	<0.001 *	1.52	1.19–1.95	0.001 *
	Richest	83.8	3.39	2.74–4.21	<0.001 *	1.46	1.00–2.12	0.045 *
Risky behaviors								
Comprehensive Knowledge	No	73.6	Ref	-	-	-	-	-
	Yes	71.9	0.92	0.63–1.35	0.663	-	-	-
Condom use	Yes	79.3	0.61	0.53–0.71	<0.001 *	-	-	-
	No	86.2	Ref			-	-	-
Number of sexual partners	1	68.0	Ref		<0.001 *	-	-	-
	2	76.8	1.56	1.28–1.89	<0.001 *	1.19	0.95–1.49	0.128
	3+	85.1	2.67	2.29–3.11	<0.001 *	1.41	1.17–1.70	<0.001 *

* *p* < 0.05.

## Data Availability

The datasets which were generated and analyzed during this current study are available in the DHS program repository via the following link: https://dhsprogram.com/data/available-datasets.cfm (accessed on 21 August 2024). The datasets are publicly available for research purposes and can be downloaded free of charge, but registration is required to ensure that your study agrees with the program’s terms and conditions.

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
