# Peer review of "Prevalence and Factors Associated with HIV Testing Among Men Aged 15–54 Years in Kenya—Evidence from the 2022 Demographic and Health Survey"

_ijerph, 2025, doi:10.3390/ijerph22081291_

Round 1
Reviewer 1 Report
Comments and Suggestions for Authors
I thank the authors for the opportunity to review their research, which determines the prevalence of HIV testing and associated factors among men aged 15 to 54 in Kenya. While the survey work is interesting overall, it presents some issues:
- The table of main characteristics (Table 1) reports the 95% confidence intervals for each variable. Why did the authors choose to include these values? I believe they are superfluous in a table describing the main characteristics of the study sample. I also believe it is important to include all study variables, both qualitative and quantitative.
- The 95% CIs and their corresponding p-values ​​are reported throughout the paper. I believe it's not necessary to report both; the 95% CI alone is sufficient.
- The 95% CI of the prevalence of subjects aged 30-34 years presents a typo (line 135), just as it does not appear to be reported for men aged between 15 and 19 years (line 136).
- In line 145, you indicated that Table 2 was inserted in the supplementary materials, but Table 1 in the supplementary materials is missing. I believe the authors are referring to Table 2 inserted in the article.
- Table 2 includes the bivariate and multivariate logistic regression analyses, but it's unclear what was calculated with the bivariate and what was calculated with multivariate analyses. Furthermore, I believe the authors performed a multivariate analysis, not a multivariato one. What type of analysis was conducted?
- Since multivariable logistic regression was conducted on numerous variables, it would be interesting to also find a type III analysis of effects in the supplementary materials.
Author Response
REVIEWER 1
Comment 1: The table of main characteristics (Table 1) reports the 95% confidence intervals for each variable. Why did the authors choose to include these values? I believe they are superfluous in a table describing the main characteristics of the study sample. I also believe it is important to include all study variables, both qualitative and quantitative.
Response: Thank you for the comment, we have now removed the 95% interval from table 1. In terms of quantitative variable, it is only the age variable for which we have now calculated the mean of 29.9 years (which we have now added to Table 1). [Page 4, line 139-140].
Comment 2: The 95% CIs and their corresponding p-values ​​are reported throughout the paper. I believe it's not necessary to report both; the 95% CI alone is sufficient.
Response: Thank you for the comment, we have now removed the p-values and only remained with the odds ratios and their corresponding 95% CI in the text. [Page 4-5, line 146-152]
Comment 3: The 95% CI of the prevalence of subjects aged 30-34 years presents a typo (line 135), just as it does not appear to be reported for men aged between 15 and 19 years (line 136).
Response: Thank you for picking the typo. Since the comparison between the 30-34 and 15-19 is reported in the next paragraph, we have now removed this statement.
Comment 4: In line 145, you indicated that Table 2 was inserted in the supplementary materials, but Table 1 in the supplementary materials is missing. I believe the authors are referring to Table 2 inserted in the article.
Response: We acknowledge that we made an error, we were referring to Table 2. The error has been corrected.
Comment 5: Table 2 includes the bivariate and multivariate logistic regression analyses, but it's unclear what was calculated with the bivariate and what was calculated with multivariate analyses. Furthermore, I believe the authors performed a multivariable analysis, not a multivariate one. What type of analysis was conducted?
Response: We agree that we performed a bivariate and multivariable analyses (not multivariate). We have now corrected this in the methods section. We conducted both bivariate analyses where the outcome is binary (whether one tested for HIV: yes/no) and each predictor variable – to get unadjusted odds ratios - and multivariable analyses with the same outcome (binary) and all the predictor variables in the final model. Bivariate analyses was conducted to determine the factors that would be included in the Multivariable model. Bivariate analyses was used to compute crude or unadjusted odds ratios (OR) and multivariable analysis was used to compute adjusted odds ratios (aOR). The odds ratios were also computed with their corresponding 95%CI’s and p-values. This has now been clarified in the Methods section of the paper.
Comment 6: Since multivariable logistic regression was conducted on numerous variables, it would be interesting to also find a type III analysis of effects in the supplementary materials.
Response: Thank you for the comment. Type III analysis tests the significance of each factor while accounting for the effects of other factors in the model. This is achieved using the test command with the appropriate syntax after fitting the svy: logistic model.
We have now performed the Type III analysis of factors by first running the svy: logistic command with all the variables in the final multivariable model and then tested each factor one by one using the test [variable]=0. The results are now presented in the Appendices. Supplementary table 1[Page 12-13, line 360-361]In the Type III analysis, most variables remained statistically significant; however, the associations with the "poorer" wealth quintile and having "two or more sexual partners" were no longer statistically significant. [Page 7, line 173-175].
Comment 7: We have noticed a high degree of similarity between the manuscript and
the published article, so we are writing to ask you to revise lines
181-185 to reduce the similarity. ("The common practice of ......
Ethiopia, and Cameroon [17].")
Response: Thank you. We have now rephrased to the following: The common practise of mandatory premarital HIV testing in many SSA countries may help explain this finding. Previous studies have shown that HIV testing is often a prerequisite for marriage in countries such as Nigeria, Ethiopia and Cameroon. [Page 7, line 187-190].
Reviewer 2 Report
Comments and Suggestions for Authors
This study utilizes 2022 DHS survey data to evaluate HIV testing coverage and associated factors among men in Kenya. While a considerable volume of research explored HIV testing coverage and its associated factors among men in various contexts (e.g., Adugna and Worku, 2022; Tetteh et al., 2021; Lakhe et al., 2020; Mandiwa and Namondwe, 2019), and specifically, findings on HIV testing coverage among Kenyan men utilizing 2022 DHS data have previously been published by Adugna and Worku (2022), the distinctive contribution of this research to the extant literature necessitates more precise articulation. Moreover, a rigorous review of the analytical methodology is warranted. Given that the DHS employs a two-stage stratified sample design, it is imperative that the analysis method accounts for this complex survey design to ensure the validity of the findings.
Specific points for improvement include:
- Line 33: Please consider incorporating information regarding the HIV prevalence rates for both males and females in Kenya.
- Lines 43-55: To enhance the specificity of the literature review, it would be beneficial to include a summary of factors previously identified as associated with HIV testing among men.
- Lines 72-73: This is questionable. There is a considerable volume of research available on HIV testing and its associated factors, particularly among men. Please re-evaluate and rephrase this statement for accuracy.
- Lines 113-120 (Analysis Method): This section requires significant clarification. Please address the following:
- Was the two-stage stratified sample design incorporated into the analysis? If so, elaborate on the specific methods used (e.g., use of survey weights, clustering, and stratification adjustments).
- What approach was used for variable selection (e.g., forward, backward, stepwise selection)?
- How many models were fitted? Were demographic, socio-economic, and sexual risk behavior factors included in the final model?
- Were two-way interaction terms explored, and if so, what were the findings?
- Line 135: The reported confidence interval appears to be incorrect. Please verify and correct this value.
- Line 170: Please discuss how the findings of this study compare with those from previous years in Kenya. Furthermore, assess whether the findings are consistent with other existing studies (e.g., Adugna and Worku, 2022; Lakhe et al., 2020; Mandiwa and Namondwe, 2019; Tetteh et al., 2021, etc.), or if there are notable discrepancies.
Author Response
Comment 1: Line 33: Please consider incorporating information regarding the HIV prevalence rates for both males and females in Kenya
Response: Despite significant progress over the past 30 years, HIV remains a major public health concern in Kenya, contributing substantially to the country’s overall disease burden. The national HIV prevalence stands at 3.7% [4], with a notable gender disparity 2.16% among males and 4.46 among females.[5]. . [Page 1, line 34-37].
Comment 2: Lines 43-55: To enhance the specificity of the literature review, it would be beneficial to include a summary of factors previously identified as associated with HIV testing among men.
Response: Papers on HIV testing among men are being referenced. [Page 2, line 45-47].
Comment 3: Lines 72-73: This is questionable. There is a considerable volume of research available on HIV testing and its associated factors, particularly among men. Please re-evaluate and rephrase this statement for accuracy.
Response: While there is a substantial body of literature on HIV testing and its associated factors, much of this research tends to focus on pregnant women, adolescent girls and young women, and key populations such as sex workers and men who have sex with men. Although studies focusing on men do exist particularly in relation to specific subgroups or comparative analyses, they are relatively fewer in number compared to those on women and other key populations. Therefore, more male-specific research is needed to address the persistent gender gap in HIV testing uptake and outcomes. [Page 2, line 72-81].
Comment 4: Lines 113-120 (Analysis Method): This section requires significant clarification. Please address the following
- Response: Was the two-stage stratified sample design incorporated into the analysis? If so, elaborate on the specific methods used (e.g., use of survey weights, clustering, and stratification adjustments).
Response: Survey weights were used to account for the survey design (weights, stratification, and clustering) since the 2022 Kenya DHS survey used the two-stage stratified sample design. [This has been added to the Methods section. Page 3, line 97-100]].
- Comment 5: What approach was used for variable selection (e.g., forward, backward, stepwise selection)?
Response: Backward elimination was used in building the final multivariable model from the bivariate analyses, where factors with p-value < 0.05 were shortlisted as candidates. [This has been added to the Methods section. Page 3, line 97-100]].
- Comment 6: How many models were fitted? Were demographic, socio-economic, and sexual risk behaviour factors included in the final model?
Response: Yes, demographic, socio-economic, and sexual risk behaviour factors were included in the bivariate analyses but only the ones that were statistically significant with P-value of less than 0.05 were included in the final model.
- Comment 7: Were two-way interaction terms explored, and if so, what were the findings?
Response: Because of parsimonious reasons, we did not include interaction terms in the final model
- Comment 8: Line 135: The reported confidence interval appears to be incorrect. Please verify and correct this value.
Response: Thank you for the comment, the error has been corrected. We have now corrected this in response to Reviewer #1 comment 2 above.
- Comment 9: Line 170: Please discuss how the findings of this study compare with those from previous years in Kenya. Furthermore, assess whether the findings are consistent with other existing studies (e.g., Adugna and Worku, 2022; Lakhe et al., 2020; Mandiwa and Namondwe, 2019; Tetteh et al., 2021, etc.), or if there are notable discrepancies.
Response: Thank you for the comment, the following paragraph have been added to the discussion section. The findings of this study, based on the 2022 KDHS, indicate a notable improvement in HIV testing uptake among Kenyan men aged 15–54 years, with 73.5% reporting ever having tested. This represents a marked increase from the 2008–09 KDHS estimate of approximately 40%[27], and a continued rise from the 2012 KAIS estimate of 55%[28], reflecting Kenya’s sustained efforts to scale up testing services through strategies such as provider-initiated testing and counselling (PITC), HIV self-testing, and community-based outreach.
These findings align with regional evidence demonstrating improved HIV testing among men following targeted interventions. For example, studies conducted in Ethiopia[7] and Senegal[29] similarly reported enhanced uptake associated with male-friendly strategies and decentralization of services. Furthermore, evidence from a sub-Saharan Africa study [21] highlighted the role of reducing stigma and improving partner communication in enhancing testing rates, consistent with Kenya’s national HIV response. Nonetheless, this study also reveals persistent disparities, particularly among rural men, those with limited education, and younger individuals, indicating the need for targeted, equity-focused interventions to ensure inclusive progress in HIV testing uptake. Page 7-8, line 177-236]].
Round 2
Reviewer 1 Report
Comments and Suggestions for Authors
I thank the authors for accepting my suggestions to improve the quality of the article, which appears clearer and more scientifically sound.
Author Response
Thank you.
Reviewer 2 Report
Comments and Suggestions for Authors
This study utilizes data from the 2022 Demographic and Health Survey (DHS) to assess the prevalence of HIV testing and its associated factors among men aged 15-54 years in Kenya. However, the prevalence of HIV testing among men in Kenya, based on the 2022 DHS data, has already been reported in the article "HIV Testing and Associated Factors Among Men (15-64 Years) in Eastern Africa: A Multilevel Analysis Using the Recent Demographic and Health Survey" (Adugna and Worku, 2022) and in the summary report of the 2022 Kenya DHS. The current study, which aims to identify factors associated with ever having undergone HIV testing, can be viewed as a subgroup analysis of the findings presented by Adugna and Worku who utilized 2022 DHS data from 11 countries in Eastern Africa. The factors associated with HIV testing in both studies appear to be quite similar. Given the current version of the manuscript, the added value of this study remains unclear, and the authors may want to consider extending beyond the existing findings.
I also have concerns regarding the analysis methods employed. As stated by the authors, the DHS study design involves two-stage, stratified, cluster sampling (lines 91-92). Survey weights were applied to account for the survey design (lines 96-98), which is crucial for estimating point estimates accurately. However, it is essential to consider the weighting, clustering, and stratification of the survey design to correctly calculate standard errors for the 95% confidence intervals of the point estimates or significance tests. The bivariable and multivariable logistic regression analyses conducted (line 123) did not account for the two-stage, cluster sampling design. The appropriate approach would be to use weighted, multi-level logistic regression or survey logistic regression, which appropriately considers the two-stage, stratified, cluster sampling design.
Author Response
Comments and Suggestions for Authors:
This study utilizes data from the 2022 Demographic and Health Survey (DHS) to assess the prevalence of HIV testing and its associated factors among men aged 15-54 years in Kenya. However, the prevalence of HIV testing among men in Kenya, based on the 2022 DHS data, has already been reported in the article "HIV Testing and Associated Factors Among Men (15-64 Years) in Eastern Africa: A Multilevel Analysis Using the Recent Demographic and Health Survey" (Adugna and Worku, 2022) and in the summary report of the 2022 Kenya DHS.
Author response: In actual fact, the data reported in the article by Adugna and Worku (2022) used old DHS data from 2014. Our article uses the most recent data from 2022 DHS data. While it is true that the 2022 DHS report has published the HIV testing prevalence for 2022, our paper is not only reporting the prevalence, it also reports the factors associated with HIV testing using the most recent 2022 Kenya DHS data for which no other paper has published.
The current study, which aims to identify factors associated with ever having undergone HIV testing, can be viewed as a subgroup analysis of the findings presented by Adugna and Worku who utilized 2022 DHS data from 11 countries in Eastern Africa.
Author response: As mentioned above, the Adugna and Worku paper used old data from 2014 DHS and our paper uses the most recent 2022 DHS data.
The factors associated with HIV testing in both studies appear to be quite similar. Given the current version of the manuscript, the added value of this study remains unclear, and the authors may want to consider extending beyond the existing findings.
Author response: Our analysis of factors associated with HIV testing are based on the most recent 2022 Kenya DHS data, whereas the factors published in the Adugna and Worku article are based on all the 11 countries. This clearly shows that our paper is different from the Adugna and Worku paper. While we appreciate the similarity in the factors, our paper reinforces similar factors based on the context of Kenya alone, as opposed to the Adugna and Worku article which is contextualized in the whole of East Africa.
I also have concerns regarding the analysis methods employed. As stated by the authors, the DHS study design involves two-stage, stratified, cluster sampling (lines 91-92). Survey weights were applied to account for the survey design (lines 96-98), which is crucial for estimating point estimates accurately. However, it is essential to consider the weighting, clustering, and stratification of the survey design to correctly calculate standard errors for the 95% confidence intervals of the point estimates or significance tests. The bivariable and multivariable logistic regression analyses conducted (line 123) did not account for the two-stage, cluster sampling design. The appropriate approach would be to use weighted, multi-level logistic regression or survey logistic regression, which appropriately considers the two-stage, stratified, cluster sampling design.
Author response: Our statistical methods specify that all our analyses were adjusted using survey weights. This adjustment accounts for the complex survey for DHS involving two-stage, stratified, cluster sampling. Our analysis adjustment using survey weights are the same as all the other papers we have published and also the same as the Adugna and Worku (2022) paper that the reviewer is referring to.